



# Glacier and Rock Glacier changes since the 1950s in the La Laguna catchment, Chile

Benjamin Aubrey Robson[1,2], Shelley MacDonell[3], Álvaro Ayala[3], Tobias Bolch[4], Pål Ringkjøb Nielsen[5], Sebastián Vivero[6]

5   1 – Department of Earth Science, University of Bergen, Norway

2 - Bjerknes Centre for Climate Research, Bergen, Norway

3 – Centro de Estudios Avanzados en Zonas Áridas (CEAZA), La Serena, Chile

– School of Geography and Sustainable Development, University of St. Andrews, UK

– Department of Geography, University of Bergen, Norway

6 – Institute of Earth Surface Dynamics, University of Lausanne, Switzerland

*Correspondence to*: Benjamin Aubrey Robson (benjamin.robson@uib.no)

## Abstract

Glaciers and rock glaciers play an important role in the hydrology of the semi-arid Northern Chile. Several studies show that glaciers have strongly lost mass in response to climate change during the last decades. The response to rock glaciers in this

region is, however, much less investigated. In this study we use a combination of historical aerial photography, stereo satellite imagery, airborne LiDAR, and the Shuttle Radar Topography Mission (SRTM) DEM to report glacier changes for the Tapado Glacier-Rock Glacier complex from the 1950s to 2020 and to report mass balances for the glacier component of the complex, Tapado Glacier. Furthermore, we examine high-resolution elevation changes and surface velocities between 2012 and 2020 for 40 rock glaciers in La Laguna catchment. Our results show how the glacier has lost $25.2 \pm 4.6\%$ of its ice

covered area between 1956 and 2020, while the mass balance of Tapado Glacier has become steadily more negative, from being approximately in balance between 1956 and 1978 ($-0.04 \pm 0.08$ m w.e. a$^{-1}$) to showing strong losses between 2015 and 2020 ($-0.32 \pm 0.08$ m w.e. a$^{-1}$). Climatological (re)-analyses reveal a general increase in air temperature, decrease in humidity, and variable precipitation since the 1980s in the region. In particular the severe droughts in the region starting in 2010 resulted in a particular negative mass balance of $-0.54 \pm 0.10$ m w.e. a$^{-1}$ between 2012 and 2015. The rock glaciers

within La Laguna catchment show heterogenous changes with some sections of landforms exhibiting pronounced elevation changes and surface velocities exceeding that of Tapado Glacier. This could be indicative of high ice contents within the landforms and also highlights the importance of considering how landforms can transition from more glacial landforms to more periglacial features under permafrost conditions. As such, we believe high-resolution (sub-metre) elevation changes and surface velocities are a useful first step for identifying ice-rich landforms.

**Keywords:** geodetic mass balance, semi-arid Andes, glacier, rock glacier, DEM, climate change



## 1.  1 Introduction

The cryosphere is in a state of rapid change, with glaciers thinning and losing mass on a global scale (Zemp et al., 2019). The Andes have some of the highest rates of glacier mass loss on the planet (Masiokas et al., 2020), with a total loss of

$-0.72 \pm 0.22$ m w.e. yr$^{-1}$ between 2000 and 2018 (Dussaillant et al., 2019; Ferri et al., 2020).

The semi-arid Andes of Chile (26–32°S) contain an assortment of glaciers and rock glaciers, and although the amount of ice is considerably less than in other South American regions, given the relative scarcity of freshwater the meltwater plays a potentially significant role in the local hydrology in the region (Pourrier et al., 2014; Schaffer et al., 2019).

Rock glaciers are lobate or tongue-shaped assemblages of ice-rich debris which slowly creep downhill due to gravity, and

are a physical manifestation of permafrost conditions (Barsch, 1996; Haeberli et al., 2006). There has been discussion about how rock glaciers form, and what constitutes a rock glacier, with two schools of thought: that rock glaciers can either be permafrost creep features derived from a talus (cryogenic)  or can be derived from glacial (glacigenic) features. Readers are directed to Berthling (2011) for an overview of the debate surrounding terminology of rock glaciers related to their origin. Some studies have recently highlighted that given permafrost conditions are present, as well as a sufficient supply of

material from the surrounding slopes, a debris-covered glacier can transition to a rock glacier (Jones et al., 2019a; Knight et al., 2019; Monnier and Kinnard, 2017). Such a transition to a more periglacial setting can have hydrological consequences as rock glaciers are more resilient to changes in climate than glaciers, and as such the frozen stores of water could be better conserved (Jones et al., 2019b; Rangecroft et al., 2013). Brenning et al. (2012) and Bolch et al. (2019) introduced the terms *ice-debris landforms* or *ice-debris complex* that include debris-covered glaciers, rock glaciers and other landforms such as

ice-cored moraines (Bolch et al., 2019; Brenning et al., 2012).

The focus of this study is a catchment containing 40 rock glaciers as well as the Tapado Glacier-Rock Glacier complex which includes a debris-covered portion of the glacier, a clean ice section, and two rock glaciers. The study region also contains the transitioning landform of Las Tetas.

Remote sensing has been used to study glacier and rock glacier changes in many parts of the world where in-situ data is

limited or non-existent (Bolch et al., 2019; Falaschi et al., 2019b; Rignot, 2002; Robson et al., 2018). Changes over extensive temporal scales can be studied by combining remote sensing datasets such as historical aerial imagery (Eriksen et al., 2018; Falaschi et al., 2019b; Kinnard et al., 2020), declassified spy datasets, namely Corona and Hexagon imagery (Bolch et al., 2011; Fariás-Barahona et al., 2020; Hanshaw and Bookhagen, 2014; Robson et al., 2018), or topographic maps (Ayala et al., 2020; Tielidze, 2016).

Recent studies have quantified glacier losses over the entire Andes (Braun et al., 2019; Dussaillant et al., 2019). Such studies provide invaluable regional scale information on glacier mass changes, however at a limited spatial resolution and temporal



scale. In particular, such analyses are incapable of studying the smaller magnitude changes in rock glaciers, which in some semi-arid catchments are more prevalent than glaciers (Azócar and Brenning, 2010; Zalazar et al., 2020).

This study focuses in detail on Tapado Glacier-Rock Glacier complex (or the Tapado complex for short), an assemblage landform consisting of a clean ice glacier (0.93 km$^2$), a debris-covered glacier (0.32 km$^2$) and a rock glacier (0.85 km$^2$). The term Tapado Glacier is used to refer to the clean ice and debris-covered ice components, but not the rock glacier. Glaciers are sparsely distributed in the semi-arid Andes, and as such studying decadal changes in surface lowering and mass balance can provide insights into the regional climate. Such analyses are hampered by a lack of in-situ glaciological data in the region. According to the World Glacier Monitoring Service, the Semi-Arid Andes has just nine glaciers that have been actively studied with in-situ mass balance measurements, only three of which (Guanaco Glacier in Chile, and Los Amarillos and Amarillo glaciers in Argentina) having records longer than 10 years (WGMS, 2020). As such, historical and recent remote sensing can be used to study glacier changes in space and time.

In this study, we present glaciers and rock glacier changes for the La Laguna catchment in northern Chile. Specifically, we focus on surface elevation changes and geodetic mass balances of Tapado Glacier between 1956 and 2020. Furthermore, we compute elevation changes between 2012–2020, as well as surface velocities for the rock glaciers in La Laguna catchment, allowing us to speculate on the ice content of individual landforms. Lastly, we compare our results with temperature, precipitation and shortwave radiation data to assess the driving forces of the observed change.

## 2. Study Site

The La Laguna catchment (~30°11'53 S, 69°56'15 W, Fig. 1) is situated at the headwaters of the Elqui River catchment, with glaciers and rock glaciers contributing between 4–13% of the annual streamflow (Favier et al., 2009; Pourrier et al., 2014; Schaffer et al., 2019). The Elqui River catchment has a total population of over 230,000 people, an agricultural production of US$40 million a year, as well as water-dependent industry such as mining (Cortés et al., 2012). La Laguna catchment lies within the Desert Andes glaciological zone of Chile which is characterised by a cold and semi-arid climate, the zone contain 188.9 km$^2$ of glaciers and rock glaciers, totalling 0.8% of the total Chilean cryosphere, of which 55% is made up of rock glaciers (Barcaza et al., 2017). The catchment is situated between ~3500 and 6216 m a.s.l. (summit of Nevado Olivares), is ~140 km$^2$ in size and contains 40 rock glaciers totalling ~9 km$^2$ in addition to a single glacier (Barcaza et al., 2017). Tapado Glacier (1.25 km$^2$) is the principal glacier in the catchment and contains a clean ice section and a debris-covered glacier section. The wider Tapado complex also includes two rock glacier tongues (0.85 km$^2$). There is evidence that the rock glaciers may pre-date the present debris-covered glacier, which together have developed a complex landform assemblage (Monnier et al., 2014). The glacier exists owing to favourable local wind and precipitation conditions due to the topography (Ginot et al., 2006; Kull et al., 2002). The ablation zone of the clean ice section of Tapado Glacier is covered by penitentes, formed by differential ablation (Sinclair and MacDonell, 2016). Borehole temperatures at the





accumulation zone revealed cold basal ice conditions, with temperatures reaching ca. -11°C at a depth of 35 m (Ginot et al., 2006).

95    Precipitation falls mainly as snow and is concentrated within the austral winter (June–September). The 0 °C MAAT isotherm is estimated to be ~4000 m asl and has been reported to be rising by 0.17°C per decade between 1974 and 2011 (Monnier et al., 2014). Ground measurements from the surface of Tapado complex showed temperatures constantly below negative at depths greater than 2 m (Monnier et al., 2014). Areas of the catchment above 4500 m asl are likely to display continuous permafrost conditions, whereas areas between 3,900 and 4,500 m asl permafrost is probably more scattered (Azocar et al.,
100   2017).

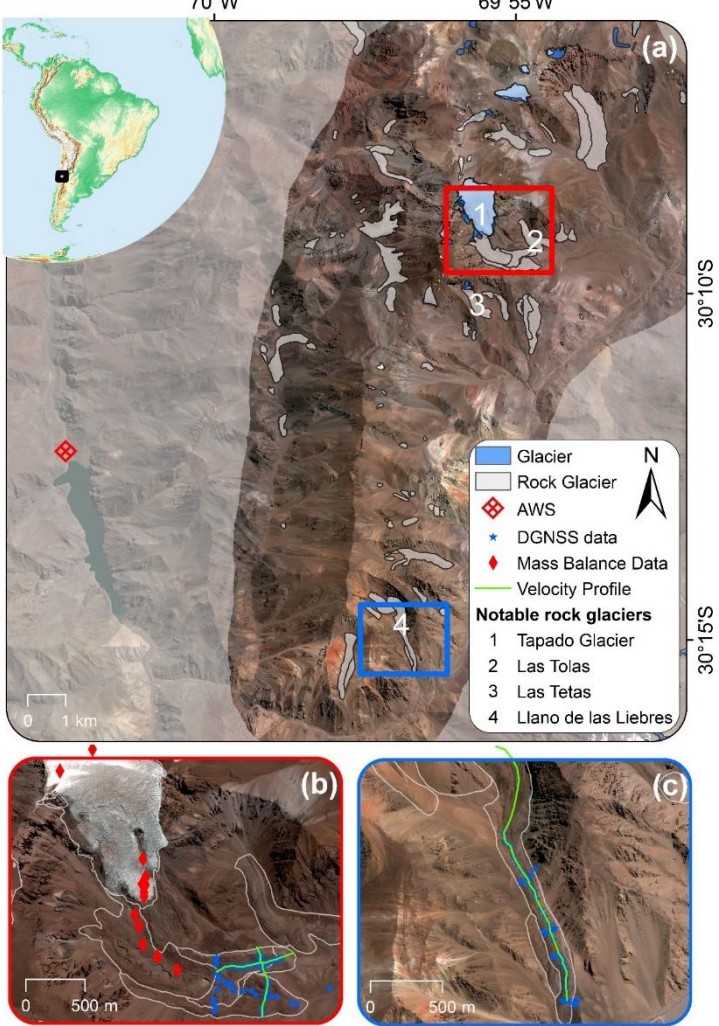

**Figure 1: Location of the La Laguna catchment and of landforms studies in this paper. Three landforms that will be discussed in the text are highlighted in (A). The Tapado complex is shown in (B) which contains a clean ice section, debris-covered glacier section, and rock glacier section. In situ field measurement locations are shown in B and C. The automatic weather station (AWS) is shown in A. Background imagery: A: Pléiades imagery (2020) overlaying Sentinel 2 *imagery (2018). B: Pléiades (2020) imagery. C: Cross-blended Hypsometric Tints obtained from naturalearthdata.com. © CNES (2018), and Airbus DS (2018), all rights reserved.***



Table 1: Datasets used in this study. For the imagery that were used to produce DEMs, the RMSE values for the Ground Control Points and Tie Points are shown. *The Landsat imagery was panchromatically sharpened from it' original 30 m spatial resolution to 15 m.

| Acquisition date | Sensor | Image IDs | Resolution (m)/Scale | Ground Control Point X-RMSE (pixels) | Ground Control Point Y-RMSE (pixels) | Tie point X-RMSE (pixels) | Tie point Y-RMSE (pixels) | Data purpose |
|---|---|---|---|---|---|---|---|---|
| 04/04/1956 | Fairchild T-11RC-10 | 26142 - 26143 | 1:70,000 | 2.32 | 2.89 | 0.15 | 0.71 | Geodetic mass balance, landform delineation |
| 31/05/1978 | RC-10 | 3945 - 3946 | 1:60,000 | 1.39 | 1.59 | 0.37 | 0.14 | Geodetic mass balance, , landform delineation |
| 11/02/2000 | SRTM | SRTM1S31W070V3 | 30 | - | - | - | - | Geodetic mass balance |
| 12/02/2000 | Landsat 7 | LE07_L1TP_233081_20000212_20170213_01_T1 | 15* | - | - | - | - | Snowline delineation, , landform delineation |
| 30/11/2010 | GeoEye-1 | 20100412144338416030311601795 | 0.5 | 1.75 | 2.73 | 0.20 | 0.04 | Geodetic mass balance, velocity calculation |
| 23/03/2012 | GeoEye-1 | 20120323144602816030311604888 | 0.5 | 0.95 | 2.17 | 0.15 | 0.12 | Geodetic mass balance, velocity calculation, , landform delineation |





| 17/03/2015 | Landsat 8 | LC08_L1TP_ 233081_2015 0317_201704 12_01_T1 | 15* | | | | | Landform delineation |
|---|---|---|---|---|---|---|---|---|
| 18/04/2015 | LiDAR | | 1 | - | - | - | - | Geodetic mass balance |
| 31/01/2019 | Pléiades | DS_PHR1A_ 20190123145 6588_FR1_P X_W070S31_ 0120 | 0.5 | 0.41 | 0.35 | 0.20 | 0.12 | Velocity calculation |
| 01/03/2020 | Pléiades | DS_PHR1B_ 20200301145 6169_FR1_P X_W070S31_ 0120 | 0.5 | - | - | 0.16 | 0.21 | Geodetic mass balance, velocity calculation, , landform delineation |

## 3. Data and materials

### 3.1 Remote sensing data

A variety of remote sensing datasets were utilised in this study (Table 1) including historical aerial photographs, airborne Light Detection and Ranging (LiDAR), and stereo satellite imagery. The LiDAR data was acquired by *Digimapas Chile* in 2015 at an original point density of 13.67 points per square metre and a vertical RMSE of 0.25 m. The data was part of a study conducted by the Chilean National Water Directorate (Dirección General de Aguas (DGA), 2015) and covers the Tapado complex and the immediate surrounding terrain. The LiDAR data was provided as a 1 m gridded DEM. Two sets of aerial photography from 1956 and 1978 were acquired by the Geographical Military Institute (IGM) of Chile. These images were provided without any calibration report, and the images themselves contains scratches and other artefacts. Additional photographs from the series which covered the entire catchment were available, however the images were of insufficient quality to process. Aerial photographs from 2000 acquired by *Servicio Aerofotogramétrico Chile* (SAF) were available, but a lack of image contrast over the accumulation area of Tapado Glacier prevented the extraction of reliable elevation values. We therefore opted to use the 30 m Shuttle Radar Topography Mission (SRTM) DEM for studying Tapado Glacier. Two Landsat images, a Landsat 7 image from 2000 and a Landsat 8 image from 2015 were used to delineate the outlines of both Tapado Glacier and the wider Tapado complex. The two scenes were panchromatically sharpened prior to analysis.



Tri-stereo satellite imagery covering the entire catchment was also used, namely two Geoeye 1 datasets (2010 and 2012) and two Pléiades datasets (2019 and 2020), both at 50 cm resolution.

### 3.2 Field data

Differential GNSS (DGNSS) data was collected as part of the ongoing monitoring program maintained by the DGA and Centre for Advanced Research in Arid Zones (CEAZA) over a period of nine years (2010 – 2019). In total there were 36

kinematic point measurements over Llano de Las Liebres rock glacier measured between 2010 and 2019 while the Tapado complex had 54 kinematic points between 2010 and 2014, four between 2010 and 2013, and ten kinematic points between 2018 and 2019. Additionally, in-situ mass balance data has been collected over Tapado Glacier. Five ablation frames were monitored between 2011–2012, eight ablation stakes were measured between 2013 and 2015.

### 3.3 Climate Data

To understand the main climatological factors that have driven the cryospheric changes observed in the La Laguna catchment, we examine meteorological records from a local Automatic Weather Station (AWS) and results from the ERA5 climate reanalysis (Muñoz-Sabater et al., 2019) in the period 1979-2020 for this area.

AWS data consists of daily average air temperature (T) and daily precipitation (P) data recorded at La Laguna AWS (Fig. 1). The AWS is maintained by the DGA and its records date back to 1978 and 1965 for T and P, respectively. We use monthly

averaged results from the ERA5 climate reanalysis extracted for the nearest point to Tapado Glacier to i) compare with the AWS data, and ii) identify changes in the main drivers of surface energy balance of glaciers in this region. In particular, we examine ERA5 data of precipitation, temperature, incoming shortwave radiation (Sin) and vapour pressure deficit (VPD). Sin and VPD are chosen due to their close relation with cloudiness and humidity, as it has been shown that these variables are linked with glacier mass balance in the Dry Andes (Kinnard et al., 2020; MacDonell et al., 2013).

**4.   Methods**

### 4.1 Generation of photogrammetric DEMs

The same workflow was broadly applied to photogrammetrically generate DEMs for 1956, 1978, 2012, and 2020. All DEMs were generated using PCI Geomatica Banff. The 2020 DEM was generated first, using the Rational Polynomial Coefficients (RPCs) provided with the imagery. No external Ground Control Points (GCPs) were used, however 1311 automatically

generated tie points were used to solve the relative orientation and perform a bundle adjustment. Epipolar images were then generated before the DEM was extracted at a 2 m posting using a Semi-Global Matching algorithm, which has been shown to outperform normalised cross-correlation and thereby produce cleaner DEMs (Hirschmüller, 2008). The DEM was then used to orthorectify the imagery and an orthomosaic was produced. The same steps were then performed for the remaining





datasets, with the 2020 orthomosaic and DEM being used as sources for GCPs in order to align the remaining datasets as
closely as possible. The residuals for the exterior (GCPs) and interior (tie points) orientations (shown in Table 1) were <3
pixels for the GCPs and <0.75 for the tie points.

## 4.2 DEM Cleaning, co-registration, and void filling

The DEMs were first cleaned using the image matching correlation scores. Additionally, following the work of Gardelle et
al. (2013), outliers were identified and removed using a value of three standard deviations of the elevation change over stable
terrain per 50 m altitudinal band.

The cleaned DEMs were co-registered together using a two-step process that was implemented using the ArcPy and Numpy
python packages. Linear co-registration biases were removed following the methods presented by Nuth and Kääb (2011)
which minimises the root mean square slope normalised elevation biases over stable terrain. The process was iterated until
the standard deviation over stable terrain changed by less than 2%. Non-linear biases were evident in the 1956 and 1978
DEMs. Third order polynomials were fitted to elevation biases on stable ground against the elevation according to the 2020
DEM, along-track, and across track following Pieczonka et al., (2013). As with the linear co-registration biases, the non-
linear corrections were iteratively applied until the improvement was less than 2%. The linear co-registration shifts are
summarised in Table 2, the non-linear co-registration shifts are shown in Fig. 2.

**Table 2: Linear co-registration shifts calculated over stable terrain between the respective DEM pairs.**

|  | Co-registration shift (m) | | |
| --- | --- | --- | --- |
| DEM pair | X | Y | Z |
| 1956 - 1978 | 0.2 | 54.2 | 16.8 |
| 1978-2000 | 1 | -6.7 | 30.2 |
| 2000 - 2012 | -1 | -0.7 | 3.1 |
| 2012 - 2015 | -2.2 | -0.9 | -8.1 |
| 2015 - 2020 | -0.2 | 1.1 | 2.6 |
| *1956 - 2020* | *0.9* | *-55.5* | *-49.9* |


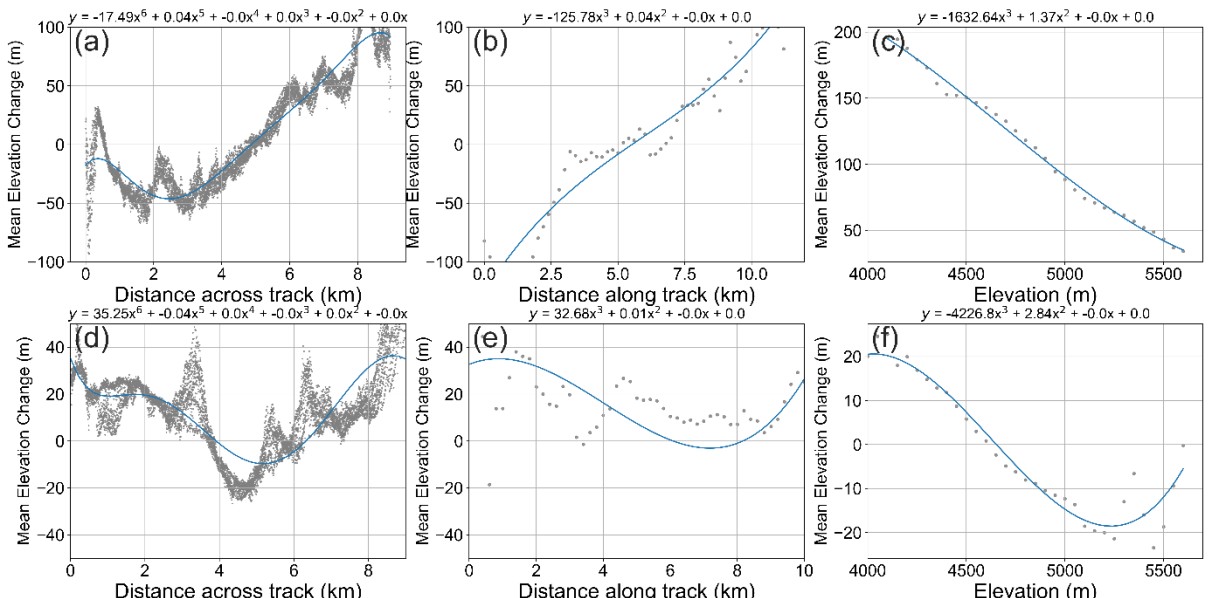

**Figure 2: Non-linear biases for the 1956 (a, b, c) and 1978 (d, e, f) DEMs when elevations are compared against the 2020 DEM over the stable (non-glacial, non-fluvial) terrain.**


After cleaning, the elevation change rasters contained small voids, predominantly on the accumulation zone of Tapado Glacier. Voids were filled by fitting a third-degree polynomial between the mean elevation and the mean elevation change per 50 m altitudinal band, a method which has been shown to outperform other void filling techniques, and can yield results suitable for studying individual glaciers (McNabb et al., 2019). To calculate the geodetic mass balance we assumed a density of glacier ice of $860 \pm 60$ kg m$^{-3}$ (Huss, 2013).

4.2.1 Radar penetration correction

The SRTM is based on C-band radar interferometry and as such has well-documented problems associated with radar penetration into the snowpack (Gardelle et al., 2012; Kääb et al., 2012). This problem is less important in the Southern Hemisphere, where the SRTM was acquired during the ablation season. A study of 20 glaciers in the Central Andes found no significant radar penetration between 3500 and ~5000 m (Farías-Barahona et al., 2019), and some studies have therefore assumed no radar penetration when using the SRTM to look at glacier changes in the Andes (Falaschi et al., 2019a; Ruiz et al., 2012). An inter-comparison between SPOT and SRTM data found however that radar penetration was occurring over the uppermost areas of the North Patagonian Icefield where wet snow and firn are present (Dussaillant et al., 2018). As such, some studies have applied linear scaled penetration corrections above the ELA, typically ranging between 0 and 5 m of penetration (Braun et al., 2019; Farías-Barahona et al., 2019; Malz et al., 2018).



Although we have aerial photographs taken within two weeks of the SRTM acquisition that could have been used as an alternative and higher-resolution elevation product, the lack of contrast on the upper reaches of the glacier prevented the extraction of reliable elevations. We therefore examined a Landsat 7 ETM+ image from the 12th of February 2000 (i.e. one day after the SRTM acquisition) and found the firn line to have a mean elevation of 5233 m a.s.l. We therefore applied a

linear scaled radar penetration from 0 m correction to 5 m correction at the highest elevation on the glacier, in line with other studies. Given the uncertainty in radar penetration we included our corrections within the uncertainty assessment.

**4.3 Elevation change uncertainty estimation**

Various methods exist for ascertaining the uncertainty of volume change estimations from DEM differencing (Magnússon et al., 2016; Nuth et al., 2012; Paul et al., 2015, 2017; Rolstad et al., 2009). We opted to follow the same methods as Falaschi et

al. (2019) who utilised similar datasets and worked in a similar setting to our study.

The uncertainty of the volume change ($E\Delta v_i$) (in m$^3$) was calculated by summing up the standard error ($E\Delta h_i$) (in m) per 50 m altitudinal band, multiplied by the area of each altitudinal band (A$_i$) in order to account for the hypsometry:

$$E\Delta v_i = \sum_i^n E\Delta h_i \times A_i \qquad (1)$$

Where the standard error ($E\Delta h_i$) is derived from the standard deviation over stable ground ($\sigma_{stable}$), divided by the effective

number of observations (N):

$$E\Delta h_i = \frac{\sigma_{stable}}{\sqrt{N}} \qquad (2)$$

N is calculated using the number of pixels (N$_{tot}$) in the DEM differencing, the pixel size (PS), and the distance of spatial autocorrection, which following Bolch et al. (2011) we took to be 20 pixels:

$$N = \frac{N_{tot} \times PS}{2d} \qquad (3)$$

For determining the total uncertainty in the geodetic mass balance of Tapado Glacier ($E\Delta v_{tot}$), we combined $E\Delta v_i$ in a square root of the sum of the squares with the uncertainties relating to the volume to mass conversion ($E_\rho$), which following Huss (2013) was taken to be ± 60 kg m$^{-3}$, the error in delineating the glacier outline ($E_a$) which was derived using the buffer method (Section 4.6), and the error related to radar penetration into snow and ice ($E^2{}_r$) when applicable:

$$E\Delta v_{tot} = \sqrt{E^2\Delta v_i + E^2{}_\rho + E^2{}_a + E^2{}_r} \qquad (4)$$

**4.4 Velocity estimation**

Surface velocities were extracted by cross-correlating features between the 2012 imagery and the 2020 Pléiades imagery. The 2012 Geoeye image covered the majority of the catchment with the exception of a small area north of the Tapado complex. As such, the velocities for this area were based on the 2010 Geoeye image. As the debris-covered portion of the Tapado complex was flowing both faster and less coherently than the rock glaciers of the catchment, the surface decorrelated

over an 8-year period, therefore features were tracked between the 2019 and 2020 Pléiades imagery. It was not possible to



track features on the clean ice section of Tapado Glacier due to deformation of the features between the images. This section was accordingly set to NoData.

Prior to performing cross-correlation it was necessary to co-register the slave images to the master images at pixel-level precision. To do this we utilised the *super-registration* routine within PCI Geomatica Banff. This routine performs a
Normalised Cross Correlation (NCC) at a given grid spacing (in our case every 5 m) to quantify and then correct any offset between orthorectified imagery. Once the imagery was co-registered, we performed the cross-correlation using IMCORR (Scambos et al., 1992), implemented through QGIS using a 64 pixel reference window and a 128 pixel search window at a spacing of 5 metres. A finer reference window could have helped define smaller scale displacements, however as the area is seismically active, and generally consists of loose, unstable rock on steep slopes, using a larger reference-window reduced
the amount of noise. The resulting displacement vector files were filtered to exclude extreme displacement values as well as erroneous flow directions and converted to annual displacements. The displacement vectors were merged into one file before an invert distance weighting (IDW) interpolation was used to create a 15-metre resolution displacement raster.

### 4.5 Surface velocity accuracy assessment

We ascertained the accuracy of our surface velocities in two ways. Firstly, we compute the standard deviation of the surface
displacements on stable ground (i.e. excluding glaciers, rock glaciers, shadows etc) where the displacement would be expected to be zero. We excluded very steep terrain (>40º) to make our accuracy assessment comparable to the terrain on the rock glaciers. It should be pointed out however that the study area mainly comprises of loose rocks and debris, and is seismically active, as such by taking the standard deviation of the stable ground as our error bars, we are most likely being too conservative. This resulted in a mean uncertainty of $\pm$ 0.03 m a$^{-1}$.
We also compare our remote sensing-based velocities to the DGNSS velocities collected from the surfaces of the Tapado complex and Llano de las Liebres rock glacier over broadly the same period.

### 4.6 Glacier and rock glacier delineation

We used the glacier and rock glacier produced by Schaffer and MacDonell, (2020) as a baseline product. This inventory was based on manual interpretation of Pléiades imagery from 2019. We manually adapted the shapefiles based on the remote
sensing data from 1956, 1978, 2000, 2012, 2015, and 2020. The Tapado Glacier-Rock Glacier complex is a consists of a clean ice glacier component, a debris-covered glacier component, and a rock glacier. As a prerequisite for deriving geodetic mass balances, we had to separate the glacier from the rock glacier. We used a combination of the surface morphology as well as the elevation change rasters for each time period as a guide for delineating glacier extent assuming that a change in surface elevation was indicative of glacier ablation or glacier dynamic processes (Kääb et al., 2014).
We used the 'buffer method' to ascertain uncertainties for landform delineation (Bolch et al., 2010; Paul et al., 2017). Buffer distances were set to half a pixel for the clean ice section, and one pixel for the debris-covered ice and rock glaciers.



## 5. Results

### 5.1 Tapado Glacier area change

Between 1956 and 2020, Tapado Glacier shrank by a total of $25.2 \pm 4.6\%$ ($0.42 \pm 0.23$ km$^2$) at a mean rate of -0.40 %a$^{-1}$ and
as of 2020 had an area of $1.25 \pm 0.01$ km$^2$ (Figure 3, Table 2). Between 1956 and 1978 the glacier area changed by $-5910 \pm$
1060 m$^2$ a$^{-1}$ ($-0.35 \pm 0.30$ %a$^{-1}$), which increased to $6818 \pm 24202$ m$^2$ a$^{-1}$ ($-0.60 \pm 2.28$ %a$^{-1}$) between 2000 and 2012. Rates of
glacier change have continued to increase, with losses of $-1.03 \pm 2.19$ % a$^{-1}$ between 2012 and 2015, although losses
decreased to $-0.16 \pm 2.77$ % a$^{-1}$ between 2015 and 2020. The clean ice section of Tapado Glacier has seen greater changes,
shrinking from $1.30 \pm 0.01$ km$^2$ in 1956 to $0.93 \pm 0.01$ km$^2$ in 2020, a reduction of $28.4 \pm 1.1\%$. The clean ice section was
largely stable between 1956 and 1978 ($-0.21 \pm 0.06$ % a$^{-1}$) with ice area changes peaking between 2012 and 2015 ($-1.63 \pm$
2.60 % a$^{-1}$). Between 2015 and 2020, the clean ice section lost $0.04 \pm 0.12$ km$^2$ of ice at a rate of $-0.82 \pm 1.70$ %a$^{-1}$.

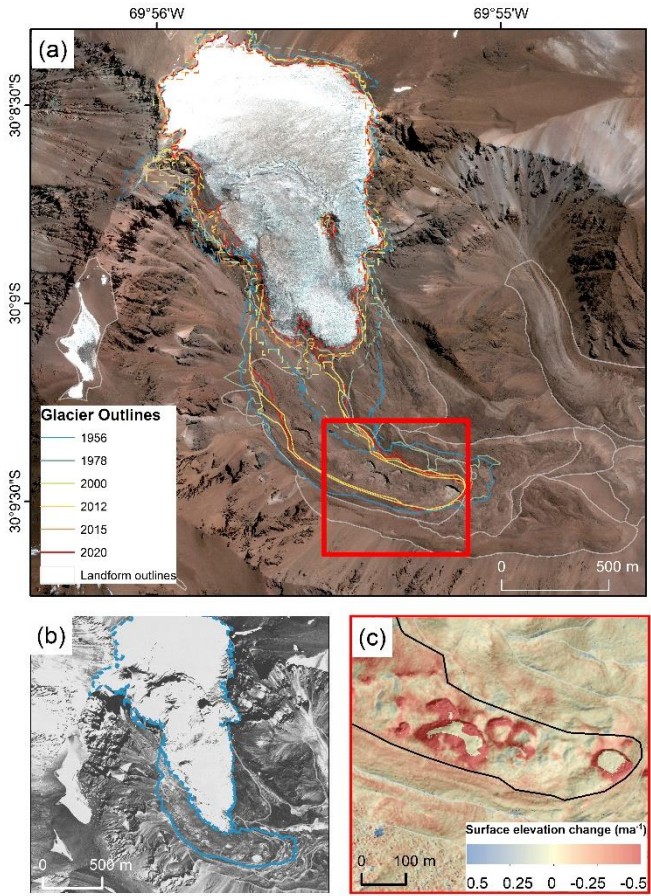

**Figure 3: (a)Change in area for Tapado Glacier; both clean ice (dashed lines) and total glacier area (solid lines), between 1956 and 2020. Background imagery is a Pléiades scene from 2020, © CNES (2018), and Airbus DS (2018), all rights reserved. (b)**
**Orthoimage and glacier outline from 1956. (c) subset of the debris-covered tongue of Tapado Glacier (outline between debris-covered ice and rock glacier shown in black) with surface elevation changes between 2012 and 2020 shown.**




## 5.2 Tapado Glacier volume changes and geodetic mass balance

Surface lowering occurred over the entire glacier (Fig. 4, Fig. 5) although the greatest rates of thinning (> 40 m in total) were found at the lower section of the clean ice at elevations of 4800–5000 m a.s.l. Elevation profiles show increasing thinning
rates in more recent time periods (). Even if thinning rates in the ablation area between 2012–2015 were ~0.1–0.3 m a$^{-1}$ more negative than previous years, thinning rates were up to -0.8 m a$^{-1}$ more negative in the accumulation area. In total Tapado Glacier has lowered by an average of -7.44 m between 1956 and 2020. The clean ice section of Tapado has been thinning at the greatest rates, with a mean annual loss of -0.64 ± 0.11 m a$^{-1}$ between 2012 and 2020 compared to -0.44± 0.11 m a$^{-1}$ for the debris-covered section of the glacier, and -0.08 ± 0.11 m a$^{-1}$ for the rock glacier section. A clear gradient in ice losses is
visible from the clean ice tongue of the glacier, to the uppermost section of the debris-covered glacier, down towards the boundary with the rock glacier. The uppermost portion of the debris-covered glacier thinned by ~0.7–0.8 m a$^{-1}$. Generally, the debris-covered glacier lost 0.3–0.5 m a$^{-1}$ although thinning rates are greatly enhanced within the vicinity of supraglacial lakes and ice cliffs, with rates of up to 2 m a$^{-1}$.

Our results show that Tapado Glacier has consistently had a negative geodetic mass balance between 1956 and 2020 (Figure
5, ) with a mean mass balance of -0.11 ± 0.05 m w.e.a$^{-1}$. The mass balance has also become steadily more negative over

**Table 3: Areas for the clean ice, debris-covered, and rock glacier components of Tapado Glacier between 1956 and 2020. Note that "total glacier" refers to the combined area of the clean ice and debris-covered ice sections.**

| Year | Area Clean Ice (km²) | Area Debris-covered Ice (km²) | Area Rock Glacier (km²) | Area Total Glacier (km²) | Area Change Clean Ice (%a⁻¹) | Area Change Debris-covered Ice (%a⁻¹) | Area Change Rock Glacier (%a⁻¹) | Area Change Total Glacier (%a⁻¹) |
|---|---|---|---|---|---|---|---|---|
| 1956 | 1.30 ± 0.01 | 0.38 ± 0.01 | 0.59 ± 0.01 | 1.68 ± 0.03 | | | | |
| 1978 | 1.24 ± 0.01 | 0.31 ± 0.01 | 0.66 ± 0.01 | 1.55 ± 0.02 | -0.21 ± 0.06 | -0.84 ± 0.54 | 0.54 ± 0.34 | -0.35 ± 0.30 |
| 2000 | 1.10 ± 0.05 | 0.30 ± 0.12 | 0.81 ± 0.16 | 1.40 ± 0.33 | -0.51 ± 0.38 | -0.15 ± 0.40 | 1.03 ± 2.03 | -0.44 ± 0.91 |
| 2012 | 1.02 ± 0.01 | 0.28 ± 0.01 | 0.86 ± 0.01 | 1.30 ± 0.01 | -0.61 ± 0.80 | -0.56 ± 4.80 | 0.51 ± 0.37 | -0.60 ± 2.28 |
| 2015 | 0.97 ± 0.04 | 0.29 ± 0.12 | 0.85 ± 0.14 | 1.26 ± 0.29 | -1.63 ± 2.60 | 1.19 ± 1.95 | -0.39 ± 0.85 | -1.03 ± 2.19 |
| 2020 | 0.93 ± 0.01 | 0.32 ± 0.01 | 0.85 ± 0.01 | 1.25 ± 0.01 | -0.82 ± 1.70 | 2.07 ± 2.84 | 0.00 ± 1.21 | -0.16 ± 2.77 |
| Total | - | - | - | - | -0.44 ± 0.01 | -0.25 ± 0.16 | 0.69 ± 0.06 | -0.40 ± 0.08 |




time, from an approximately in balance mass balance of -0.04 ± 0.08 m w.e.a⁻¹ between 1956 and 1978 and -0.03 ± 0.39 m w.e.a⁻¹ between 1978 and 2000. The glacier mass balance was most negative between 2012 to 2015 (-0.54 ± 0.10 m w.e.a⁻¹). Between 2015 and 2020 the mass balance was less negative in the most recent years, with a value of -0.32 ± 0.08 m w.e.a⁻¹.

**Table 4: Summary of total glacier (clean ice and debris-covered ice) area changes, mean surface change, and geodetic mass balance for Tapado Glacier between 1956 and 2020.**

| Time period | Mean Surface Change (ma⁻¹) | Mass balance (m w.e. a⁻¹) | Mean Surface Change (Clean glacier only) (ma⁻¹) | Mass balance (Clean glacier only) (m w.e. a⁻¹) | Volume change (m³a⁻¹) |
|---|---|---|---|---|---|
| 1956 - 1978 | -0.05 ± 0.09 | -0.04 ± 0.08 | -0.16 ± 0.14 | -0.14 ± 0.18 | -61,928 ± 111,471 |
| 1978 - 2000 | -0.05 ± 0.46 | -0.03 ±0.39 | -0.41 ± 0.62 | -0.35 ± 0.52 | -55,245 ± 508,257 |
| 2000 – 2012 | -0.21 ± 0.18 | -0.17 ± 0.15 | -0.23 ± 0.99 | -0.19 ± 0.84 | -214,387 ± 183,760 |
| 2012 – 2015 | -0.64 ± 0.11 | -0.54 ± 0.10 | -1.21 ± 0.21 | -1.03 ± 0.18 | -617,680 ± 106,164 |
| 2015 – 2020 | -0.38 ± 0.08 | -0.32 ± 0.08 | -0.43 ± 0.11 | -0.37 ± 0.11 | -351,891 ± 74,082 |
| *1956 - 2020* | *-0.08 ± 0.05* | *-0.11 ± 0.05* | *-0.20 ± 0.10* | *-0.15 ± 0.08* | *-74,082 ± 46,301* |






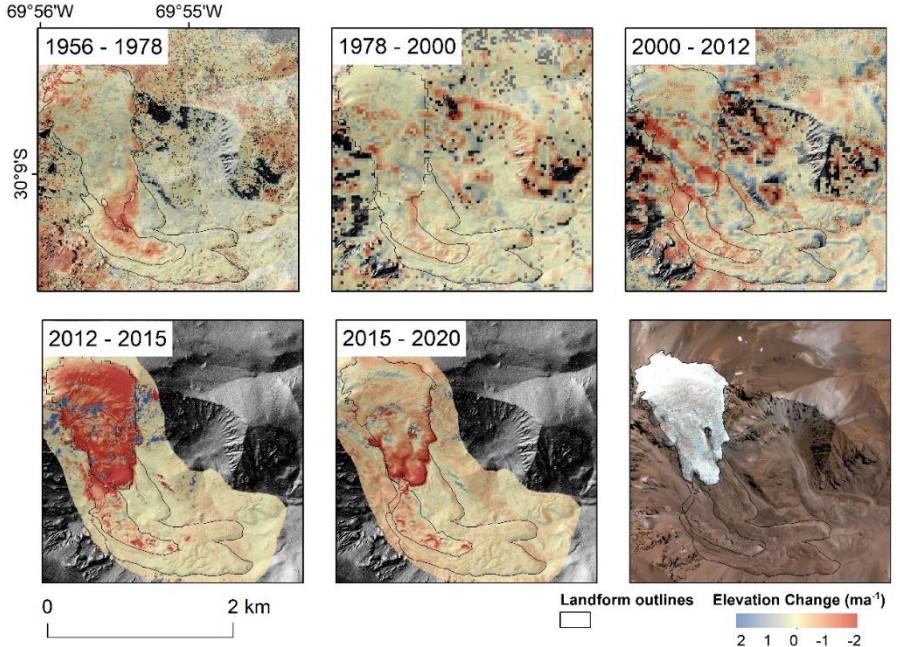

**Figure 5: Mean annual surface elevation changes for Tapado Glacier between 1956 and 2020. The outline for the Tapado Glacier-Rock Glacier complex are shown, along with subdivisions for the clean ice section, debris-covered ice section, and rock glacier section.**

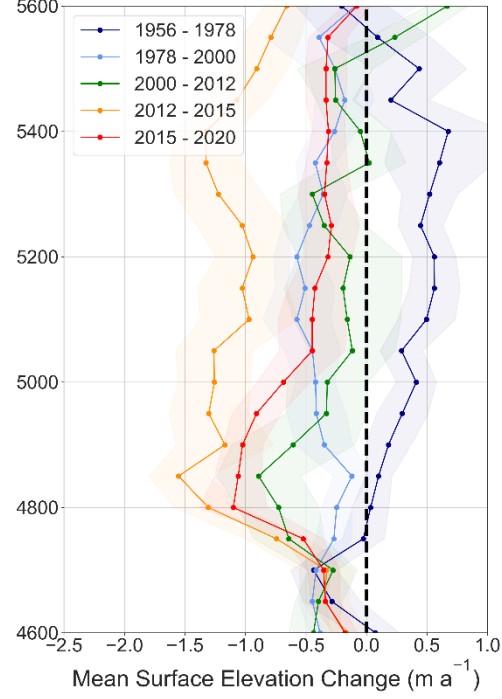

**Figure 4: Change in mean surface elevation per year per altitudinal band for the different time periods for Tapado Glacier. Shaded error bars represent the standard deviation at each elevation band.**



### 5.3  La Laguna catchment rock glacier changes

Of the 40 rock glaciers within La Laguna catchment, we had data to study changes of 35 features. Surface elevation changes between 2012 and 2020 reveal heterogenous changes (Table 4, Figure 6). Most of the rock glaciers show clear patterns of
alternating traverse bands of elevation gains and elevation losses related to rock glacier flow (Fig. 5). The lobal fronts of the landforms generally show increases in elevation associated with the gradual advance and the accumulation of rocky debris. Additionally, some of the rock glaciers exhibit longitudinal bands of elevation gain and loss.

Fourteen of the 35 rock glaciers did not show any significant change (i.e. greater than the uncertainty) over the eight-year period. Additionally, 23 of the landforms underwent surface lowerings less than 0.03 ma$^{-1}$. Three rock glaciers showed
modest gains in elevation which we interpret to be a result of the accumulation of debris. These rock glaciers are all found in narrow valleys where material from the surrounding slopes could avalanche onto the surface of the landforms. Several of the rock glaciers exhibit modest surface lowerings rates as high as > 0.1 ma$^{-1}$ between 2012 and 2020. One of these (Llano de Las Liebres; ID CL104300007; Fig. 5H) has an extensive area of surface lowering near the mid-point of the rock glacier where the surface dropped by ~2–3 m in total. The lowermost section of the landform showed drops in elevation of up to 1 m
in total between 2012 and 2020, while the uppermost stretches showed little to no change. Another feature (ID CL104300032, Las Telas) had a mean lowering of -0.11 ± 0.05 ma$^{-1}$. Upon closer inspection this landform seems to be a debris-covered glacier transitioning into a rock glacier, with both glacial and periglacial characteristics visible on the surface (Fig. 12).

**Table 5: Mean surface velocities and mean surface elevation change for ice-debris landforms within La Laguna catchment between 2012 and 2020.**

| Landform ID | Name (if available) | Mean Surface Elevation Change (ma$^{-1}$) | Mean Surface Velocity (ma$^{-1}$) |
|---|---|---|---|
| CL104300001 | | -0.04 ± 0.01 | 0.02 ± 0.03 |
| CL104300002 | | 0.03 ± 0.01 | 0.03 ± 0.03 |
| CL104300003 | | n/a | 0.05 ± 0.03 |
| CL104300004 | | n/a | 0.10 ± 0.03 |
| CL104300006 | | 0.03 ± 0.03 | 0.23 ± 0.03 |
| CL104300007 | Llano de las Liebres | -0.01 ± 0.01 | 0.36 ± 0.03 |
| CL104300008 | | 0.00 ± 0.02 | 0.15 ± 0.03 |
| CL104300009A | | -0.01 ± 0.04 | 0.19 ± 0.03 |
| CL104300009B | | -0.01 ± 0.01 | 0.51 ± 0.03 |
| CL104300010 | | 0.00 ± 0.02 | 0.27 ± 0.03 |
| CL104300011 | | 0.02 ± 0.03 | 0.14 ± 0.03 |
| CL104300012 | | -0.06 ± 0.05 | 0.07 ± 0.03 |





| | | | |
|---|---|---|---|
| CL104300013 | | -0.03 ± 0.05 | 0.03 ± 0.03 |
| CL104300014 | | 0.02 ± 0.01 | 0.14 ± 0.03 |
| CL104300015 | | 0.09 ± 0.02 | 0.03 ± 0.03 |
| CL104300017 | | 0.03 ± 0.04 | 0.02 ± 0.03 |
| CL104300018 | | -0.02 ± 0.05 | 0.08 ± 0.03 |
| CL104300020 | | -0.02 ± 0.02 | 0.20 ± 0.03 |
| CL104300021 | | 0.03 ± 0.06 | 0.21 ± 0.03 |
| CL104300030 | | 0.02 ±0.02 | 0.04 ± 0.03 |
| CL104300032 | Las Tetas | -0.11 ± 0.05 | 0.61 ± 0.03 |
| CL104300036 | | 0.03 ± 0.01 | 0.14 ± 0.03 |
| CL104300037A | | -0.01 ± 0.03 | 0.11 ± 0.03 |
| CL104300037B | | 0.00 ± 0.02 | 0.12 ± 0.03 |
| CL104300039A | Tapado Debris-Covered Glacier | -0.35 ± 0.02 | 0.28 ± 0.03 |
| CL104300039B | Tapado Rock glacier | -0.07 ± 0.01 | 0.83 ± 0.03 |
| CL104300039C | Tapado Rock Glacier | -0.05 ± 0.04 | 0.96 ± 0.03 |
| CL104300041 | Las Tetas | -0.13 ± 0.01 | 0.35 ± 0.03 |
| CL104300042 | | -0.13 ± 0.03 | 0.24 ± 0.03 |
| CL104300053 | | 0.03 ± 0.07 | 0.30 ± 0.03 |
| CL104300054 | | n/a | 0.78 ± 0.03 |
| CL104300058 | | n/a | 0.07 ± 0.03 |
| CL104300059 | | n/a | n/a |
| CL104301020A | | -0.02 ± 0.02 | 0.55 ± 0.03 |
| CL104301020B | | 0.01 ± 0.05 | 0.84 ± 0.03 |
| CL104301021 | | -0.05 ± 0.01 | 0.51 ± 0.03 |
| CL104301022 | | -0.05 ± 0.01 | 0.34 ± 0.03 |
| CL104301023 | | -0.05 ± 0.05 | 0.51 ± 0.03 |
| CL104301024 | | 0.01 ±0.02 | -0.11 ± 0.03 |
| CL104301025 | | -0.03 ± 0.06 | 0.06 ± 0.03 |



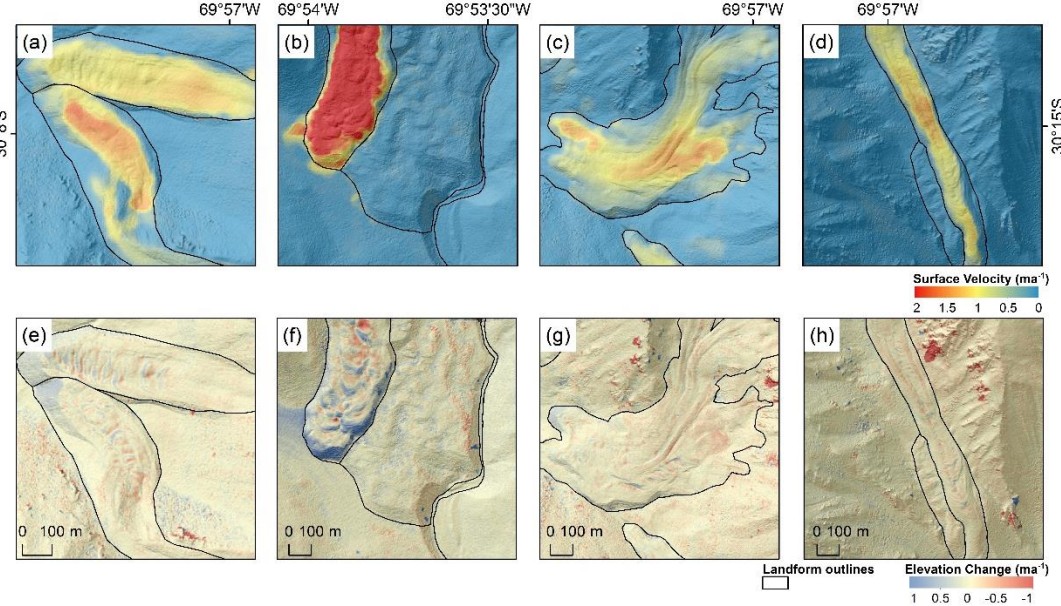

**Figure 6: Mean annual surface velocities and surface elevation changes between 2012 and 2020 for four select ice-debris**
**landforms. The landforms shown are CL104301020A and CL104301020B (in A and E), CL104300054 (shown in B and F),**
**CL104301021 (shown in C and G) and Llano de Las Liebres (CL104300007) (shown in D and H).**

### 5.4 La Laguna catchment rock glacier surface velocities

On average, the rock glaciers within the catchment have been flowing at a rate of $0.54 \pm 0.03$ m a$^{-1}$ between 2012 and 2020.
The surface velocities of landforms varied greatly within the catchment, from landforms that were inactive (moving at rates
below the uncertainty, $<0.03 \pm 0.03$ ma$^{-1}$) to those flowing at mean velocities $>0.9 \pm 0.03$ m a$^{-1}$. Although landform in the
catchment with the highest mean displacement was the debris-covered portion of the Tapado complex ($1.1 \pm 0.03$ ma$^{-1}$)
several rock glaciers are flowing at comparable rates such as CL104301020 ($0.96 \pm 0.05$ma$^{-1}$) and CL104300054 ($0.89 \pm$
$0.05$ma$^{-1}$). Additionally, in some areas of individual rock glaciers (such as CL104300054), velocities of up to 4 m a$^{-1}$ were
observed. Five of the 40 rock glaciers had either no observed velocity, or an observed velocity less than the detection level of
$0.03$ m a$^{-1}$.

Several rock glaciers have distinct active and inactive areas. For example, CL104300054 (shown in Fig. 6b) has a ~250 m
wide section with surface velocities flowing up to 4 m a$^{-1}$ on the western side of the landform, while the eastern section is
largely inactive with no discernible velocity. Similar heterogenous velocities are also seen in CL104300038 and
CL104301022. On the Tapado complex and to an extent, Las Tetas, surface velocities are noticeably higher downstream of
ponds on the surface.





## 5.5 Comparison with field data

When the velocities computed from optical feature tracking were compared with the DGNSS derived displacements (Figure 7) there was a strong positive relationship ($R^2$ = 0.81) although there is noticeable bias, with the feature tracking

underestimating the velocity according to the DGNSS data by an average of 0.16 m a$^{-1}$. There are several points however where the DGNSS data shows velocities an order of magnitude faster than measured with the feature tracking. These points seemed to occur in two settings, either on the margins of the rock glaciers, or within the vicinity of supraglacial lakes and ice cliffs on the Tapado complex. In both cases, we speculate that the spatial resolution of the feature tracking dataset cannot resolve such contrasts in velocity over a small area, resulting in an underestimation of surface velocity when compared to the

individual boulders tracked with the DGNSS.

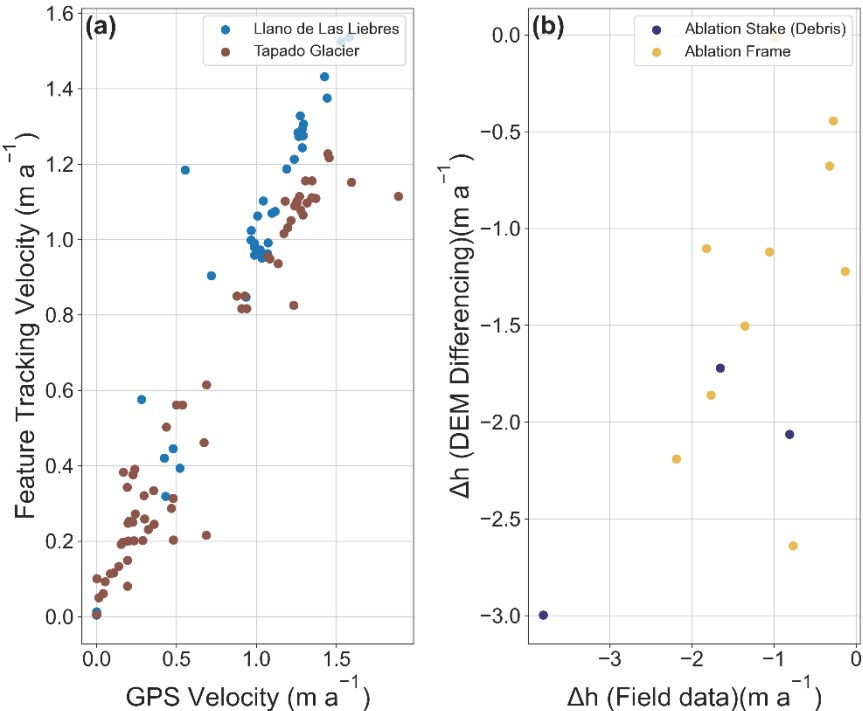

**Figure 7: (A) comparison of surface velocities derived from optical feature tracking with displacements measured with a DGNSS. (B) comparison between surface elevation changes between 2012 and 2015 derived from remote sensing, and surface elevation change measured at ablation stakes between 2011 and 2012, and 2014 and 2015.**

We also compared the optical feature tracking data to the DGNSS data along a longitudinal and traverse profile on the Tapado complex, as well as one longitudinal profile on Llano de Las Liebres rock glacier (Figure 8). Similar patterns between both database sets were observed, with mean deviations for Tapado Profile 1, Tapado Profile 2, and Llano de las Liebres of 0.9 m a$^{-1}$, 0.07 m a$^{-1}$, and 0.02 m a$^{-1}$ respectively. It should be noted that while the Llano de Las Liebres DGNSS





data spans approximately the same time period as the remote sensing data (2010–2019 vs 2012–2020), the DGNSS data

collected on the Tapado complex is from a range of dates within that time interval. Given that several of the measurements were taken on unstable areas close to supraglacial ponds and undercutting ice cliffs, the timing of the measurements may be important, perhaps partially explaining the larger variations between the methods.

We also compared the 13 mass balance measurements against surface elevation data between 2012–2015 (Fig. 8B). Given the range of years reflected in the field data, we only make visual comparisons of the relationship between the field data and

the remote sensing analyses, which appears to be a weak yet positive. Note that 10 of the 13 measurements were taken on the part of the clean glacier where penitentes are found as, which are likely to induce further errors into the approach (similar to how ablation is over- or under-estimated using point measurements in penitente fields). Three of the four largest outliers on Fig. 8B were from the penitentes while one point was from the debris-covered glacier.

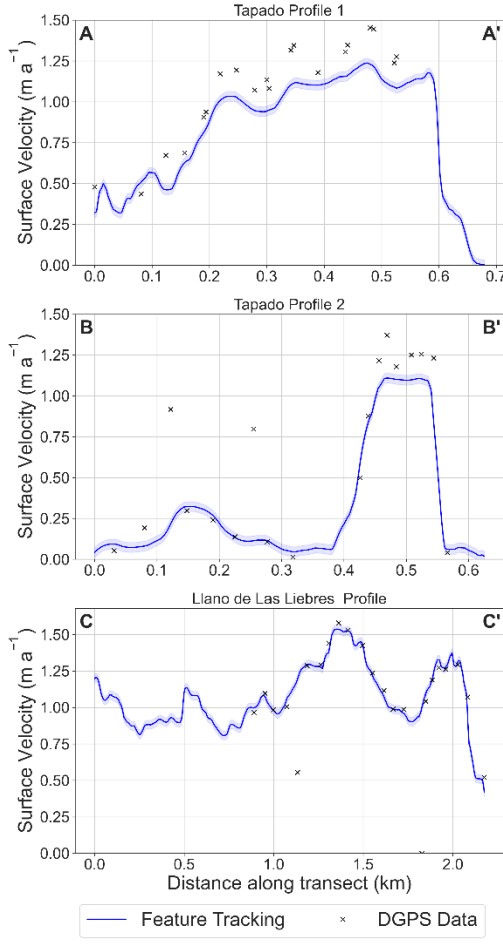

**Figure 8: Comparison of optical feature tracking based velocities and DGNSS derived velocities along three profiles on the Tapado complex and Llano de Las Liebres. The locations of these transects are shown on Fig. 1. The blue shading represents the uncertainty of the feature tracking calculated based on the stable terrain.**





## 5.6 Climatological context

We analysed air temperature (T) and precipitation (P) data recorded at La Laguna AWS (shown in Fig. 1). T shows an
increasing trend in the period 1980 to 2000 (blue line, Fig. 9a), but there has not been any notable increase subsequently.
Precipitation shows a more complicated temporal variability with various humid and dry periods (blue line, Fig. 9b). The
period 2009-2014 was one of the driest periods on record and corresponds to a severe drought in central Chile,
unprecedented in extent and duration (Garreaud et al., 2017). Decadal averages of mean annual T and total annual P during
the 2010's (2010-2019) were 0.65°C higher and 90 mm yr-1 lower than in the 1980's (1980-1989), respectively. These local
data compare very well in terms of temporal variability with monthly averaged results from the ERA5 climate reanalysis
(orange lines, Fig. 9a-b). However, exact values present some differences (not shown) that are likely related with elevation
differences between the AWS and the grid point.

An increase of Sin and VPD, derived from ERA5, suggest that cloudiness and humidity have decreased since the 1980's.
Fig. 9c shows that decadal averages of mean annual Sin over the 2010's increased by 2.7 Wm$^{-2}$ relative to the 1980's (~1%
increase) (Fig. 9c). Mean annual values of Sin as low as 280 Wm$^{-2}$ are estimated for the 1980's but are not repeated after the
1990's. Decadal averages of mean annual VPD over the 2010's increased by 48 Pa relative to the 1980's (~20% increase)
(Fig. 9d).

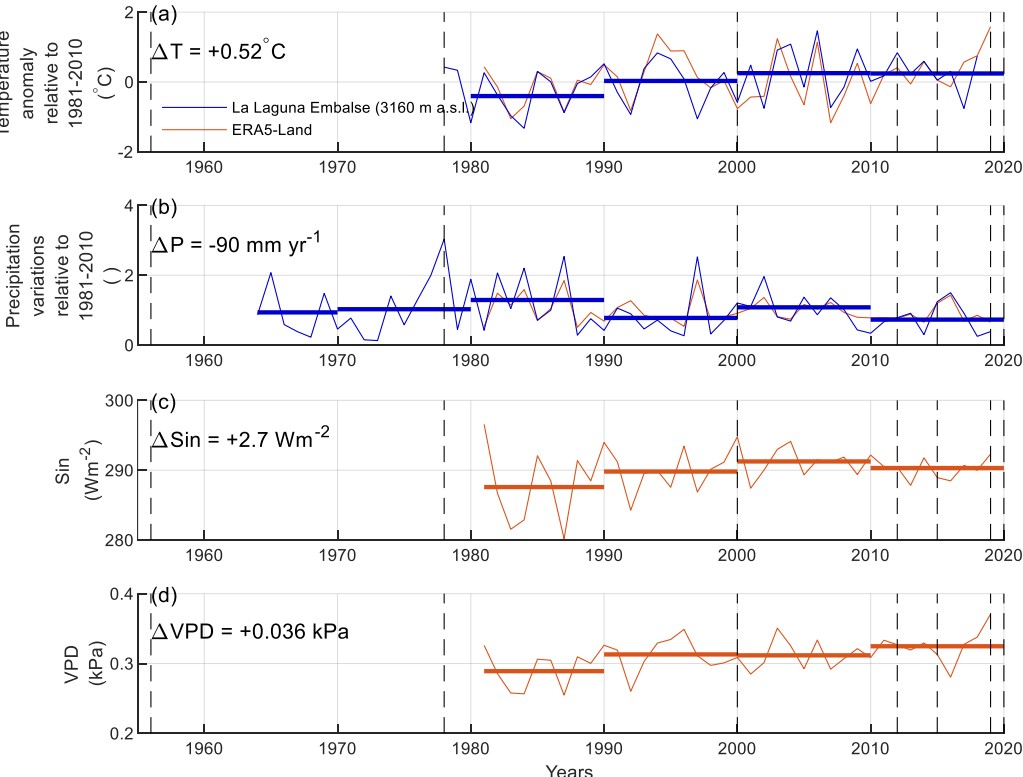

**Figure 9: Annual time series of (a) mean annual air temperature, (b) total annual precipitation, (c) mean annual incoming shortwave radiation (Sin), and (d) vapour pressure deficit (VPD) are shown in thin continuous lines. Decadal averages are shown in thick continuous horizontal lines. While (a) and (b) are variables recorded at La Laguna station, (c) and (d) are variables extracted from the ERA5-Land climate reanalysis. Vertical dashed black lines correspond to the years at which Tapado Glacier DEMs are available. Δ values correspond to differences between decadal averages over the 2010's and the 1980's.**



## 6. Discussion

### 6.1 Comparison with regional glaciological studies

Tapado Glacier has been shown to have undergone no noticeable change in area between 2000 and 2015 by Barcaza et al. (2017), although unlike this study, the glacier and rock glacier components were not considered separately. Hess et al. (2020) studied the glaciers within the Huasco Valley, in the region to the north of La Laguna, and found the glaciers were shrinking at an average rate of 2.2% a$^{-1}$ between 2000 and 2015, which although exceeding even the highest retreat rate measured at Tapado Glacier of $1.03 \pm 2.19\%$ a$^{-1}$ between 2012 and 2015, is within the error margin.

We compared our results with those of Braun *et al.*, 2019 and Dussaillant *et al.,* 2019 who studied glacier changes over the entire Andes between 2000 - 2015 and 2000 - 2018, respectively. As the former of these studies included only the clean ice part of Tapado Glacier ,we also exclude the debris-covered ice from out comparison (Fig. 10). The former used repeat bi-static synthetic aperture radar interferometry, while the latter used ASTER optical stereo imagery. Our results agree well with Dussaillant *et al.* (2019), although the datasets diverge slightly at elevations greater than 5400 m a.s.l. In addition to the differing spatial resolutions of the results, an additional reason for this disparity could be the greater radiometric resolution of the Pléiades sensor (12 bit) vs ASTER (8 bit) allowing improved image contrast within the accumulation area.

The elevation changes over Tapado Glacier according to Braun *et* al. (2019) are noticeably more positive than either our results or the results from Dussaillant *et al.* 2019. Although a similar elevation profile pattern can be observed between all three datasets, Braun *et* al. (2019) shows glacier thickening in elevations above 4910 m a.s.l and at most elevations is ~ 1 m a$^{-1}$ more positive than the results from either this study or from Dussaillant *et al.* 2019. Additionally, the spread of data is wider, the mean standard deviation of elevation change over the entire glacier is 0.26 m for Dussaillant *et al.* 2019, 0.41 m for this study, and 1.58 m for Braun *et* al. (2019). From this we can state that our results conform to the regional analysis performed by Dussaillant *et al.* 2019, while the InSAR based approach of Braun *et al.,* (2019) underestimates glacier mass losses.

Our geodetic mass balance for Tapado Glacier of $-0.24 \pm 0.10$ m w.e.a$^{-1}$ between 2000 and 2020 compared very well with the regional average for the Dry Andes of $-0.28 \pm 0.18$ m w.e. a$^{-1}$ between 2000 and 2018 according to Dussaillant *et al.* 2019, and it is more negative than the value of $0.03 \pm 0.17$ m w.e. a$^{-1}$ given by Braun *et al.* (2019)





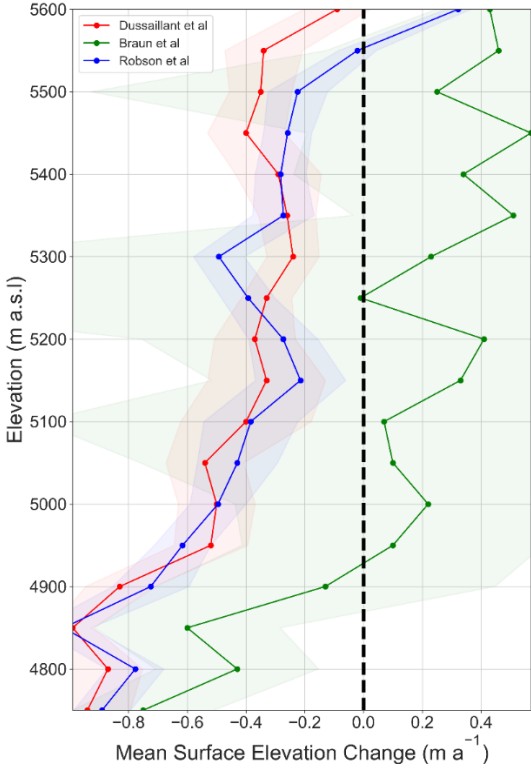

**Figure 10: Comparison of mean surface elevation change per elevation bin for the clean ice of Tapado Glacier between this study, and the results (blue) against Braun *et al*., 2019 (green) and Dussaillant *et al.,* 2019 (red). Shaded error bars represent one standard deviation of elevation changes per elevation bin.**

**6.2 Comparison with other mass balance studies**

Other studies have utilised remote sensing archived to study glacier changes over longer time spans, although comparisons with our study are not always straightforward due to differing climatic settings. The only other glacier within the Dry Andes that has been investigated over such temporal scales is Guanaco Glacier, approximately 90 km to the north of Tapado Glacier (Kinnard et al., 2020) . Guanaco Glacier had a negative balance of -0.15 ± 0.07 m w.e.a$^{-1}$ between 1955 and 1978, which is slightly more negative than Tapado Glacier's value over the same period of -0.04 ± 0.08 m w.e.a$^{-1}$. Although the

other time intervals for the same glacier do not match those from this study, both Guanaco and Tapado Glaciers are shown to have a progressively more negative mass balance in recent years, Guanaco's mass balance of -0.40 ± 0.04 m w.e a$^{-1}$ between 2005–2011 compared to Tapado Glaciers's mass balance of -0.21 ± 0.18 between 2000 and 2012. Similarly, Guanaco had a mass balance of -0.81 ± 0.09 m w.e a$^{-1}$ between 2011 and 2015 (measured using the glaciological method) compared with Tapado Glacier's value of -0.64 ± 0.11 m w.e a$^{-1}$ between 2012 and 2015.

Our results show that since the 1950s, Tapado Glacier has had a geodetic mass balance approximately in line with the glaciers within the Maipo Basin of Central Chile (~400 km to the south) which had a regional average -0.12 ± 0.06 m w.e.a$^{-1}$





with individual glacier mass balances ranging from -0.28 ± 0.07 to 0.18 ± 0.07 m w.e.a⁻¹ (Farías-Barahona et al., 2020), but is less negative than El Morado Glacier, also within the Maipo Basin, (-0.39 ± 0.15 m w.e.a⁻¹ between 1955 and 2015) although this glacier is situated at a lower elevation than Tapado Glacier and has a calving front (Fariás-Barahona et al.,

2020). Tapado Glacier also has a less negative mass than the Patagonian glaciers of Río Oro (-0.78 ± 0.11), Río Lácteo (-1.59 ± 0.08), and San Lorenzo Sur (-1.65 ± 0.07) measured between 1958 - 2018 (Falaschi et al., 2019b).

**6.3 Comparison of rock glacier results**

Comparatively few studies have investigated rock glacier changes in the region. Dos Lenguas rock glacier in the Argentinian semi-arid Andes was investigated using both repeat Sentinel-1 interferometry (Strozzi et al., 2020) and UAV-based feature

tracking (Halla et al., 2020). Rates of surface velocities were found to range from 1.5–2 m a⁻¹ which are on the higher end of the rates we observed in La Laguna catchment. Similarly, rock glaciers and ice-debris complexes in the Cordón del Plata region of Argentina were found, for a similar period, to have higher rates of deformation with average surface velocities of up to 2 m a⁻¹ and maximum velocities of up to 6 m a⁻¹ (Blöthe et al., 2018, 2020). Monnier and Kinnard (2017) found surface elevations ranging between $0.50$ and $1.10$ m yr⁻¹ over three rock glaciers in the central Andes. One of these landforms,

Las Tetas, situated 1.5 km to the south of the Tapado complex, was measured to have a mean velocity of 0.86 m a⁻¹ between 2000 and 2012, a little higher than the value of 0.69 m a⁻¹ that we measured for the same landform. Blöthe et al., (2020) also reported that rock glacier velocities varied within individual landforms, with high velocities often focused in narrow belts.

**6.4 Implications for hydrological resources**

Our study has revealed heterogeneous patterns of cryospheric activity across the La Laguna catchment which can have

implications for understanding the hydrological significance of the catchment's landforms. In particular, we found that certain rock glaciers had sections that were flowing at rates in excess of 4 m a⁻¹, considerably faster than the catchment average of 0.54 ± 0.03 m a⁻¹, as well as the surface velocities observed on the Tapado complex and Las Tetas rock glacier which are at least partly glacial in origin. Additionally, some rock glaciers had areas that had undergone elevation changes of several metres between 2012 and 2020. Both high surface velocities and decreases in elevation can be indicative of active ice

within landforms with Halla et al. (2020) suggesting that active flow was indicative of ice-rich permafrost situated above the bedrock. Although seasonal and multi-annual rock glacier velocities can be attributed to variations in water input from precipitation and snow melt (Cicoira et al., 2019), such variations are not statistically significant over the eight year temporal baselines used in this study to derive velocities.  Monnier et al (2014) performed Ground Penetrating Radar (GPR) surveys across the Tapado complex and found evidence for massive ice not only within the debris-covered section of the complex,

but also within the rock glacier components, which likely explains the pockets of surface elevation change we observe across several rock glaciers in the catchment. We find that for transitional landforms, although surface elevation changes can be an order of magnitude larger on debris-covered glacier components, the highest surface velocities of a landform are generally



found at the lowermost portions of rock glaciers. We interpret this as to suggest that ice lenses are found throughout the landforms. We believe a combination of surface elevation changes and surface velocity data can be used to identify the rock

glaciers in the region with the largest ice volumes that could be validated with geophysical techniques such as GPR or electrical resistivity tomography (ERT).

## 6.5 Transitional landforms

Our study site contains one landform that is transitioning from a debris-covered glacier to a rock glacier (Las Tetas) and one where glacial and periglacial components are interacting (Tapado Glacier-Rock Glacier complex). A transformation in

geomorphology is visible on the surfaces of both Las Tetas, and to a lesser extent, the Tapado complex (Fig. 11). Many of the surface ponds and pronounced chaotic glacial geomorphology due to unstable surfaces and spatially incoherent surface change visible on the 1978 image have been replaced by geomorphology indicative of coherent, periglacial creep. Other studies have highlighted contrasting patterns between rock glacier components and debris-covered glacier components of landforms (Blöthe et al., 2018; Monnier and Kinnard, 2017). A distinction can be seen on the surface of the Tapado complex

in both surface elevation changes and velocity datasets that correspond approximately to the geomorphological units depicted by Monnier et al. (2014). A similar, if smaller, magnitude distinction in landform components is visible on Las Tetas. In both cases, higher velocities and larger surface elevation changes are observed within the vicinity of surface ponds, and both landforms have faster flowing rock glacier tongues. In particular, the northern tongue of the Tapado complex is flowing at rates of > 1.1 m a$^{-1}$. The border between rock glacier and debris-covered glacier geomorphology is observed to

have moved approximately 150 m up glacier between 1956–1978 and 2015–2020 on the Tapado complex. Monnier and Kinnard (2017) reported similar changes for Las Telas, with a decrease in the amount of surface ponds, and more coherent velocities. Jones et al. (2019) suggest that the supply of sediment, in particular rock falls, from the surrounding terrain is a key driver in the transition from a debris-covered glacier to a rock glacier. The rock glacier part of the Tapado complex has a source of sediment from the moraine deposits, the debris-covered glacier, as well as rock falls.




**Figure 11: Comparison of orthoimagery from 1978 (aerial photography) and 2020 (Pléiades imagery) of the surfaces of the Tapado complex (red frame; A, B, C) and Las Telas (blue frame; D, E, F) as well as changes in the surface elevation between 1978 and 2020. The respective outlines from 1978 and 2020 are shown. © CNES (2018), and Airbus DS (2018), all rights reserved.**





## 6.6 Climatological context

While the increase of air temperature can be understood within global warming trends, the decrease of precipitation, cloudiness and humidity since the 1980's is likely associated with the southern migration of the South Pacific Subtropical High (Flores-Aqueveque et al., 2020). A link between this migration and negative glacier mass balance due to scarce precipitation in this region was identified by Kinnard et al. (2020). In the case of Tapado Glacier, the warmer and drier conditions have likely favoured melt on the penitentes and debris-covered areas (below 5000 m a.s.l.), and sublimation on

the high-elevation, penitente-free areas (above 5000 m a.s.l.) (Ayala et al., 2017; MacDonell et al., 2013). Although some relatively humid periods and years might have produced occasional mass gains, the overall warming and drying trends are consistent with the negative geodetic mass balance obtained in this study. Unfortunately, due to the lack of data it is not possible to analyse the period before 1979 (except for precipitation at La Laguna station that dates back to 1965). Although we evaluated other climate reanalysis for that period (including ERA5 back extension) we discarded them because their

associated uncertainty is larger than that from the reanalyses focusing on the period after 1979 (beginning of the satellite era).

Central and Northern Chile saw a series of severe droughts between 2010 and 2015 with annual mean reductions in precipitations as much as 45% (Garreaud et al., 2017). This drought corresponds with the most negative period of glacier mass balance we observed at Tapado Glacier between 2012 and 2015 (-0.54 ± 0.10 m w.e. a$^{-1}$). Similar glacier responses to

the drought have been reported on other glaciers in Central and North Chile (Farías-Barahona et al., 2019, 2020; Masiokas et al., 2020). This connection suggests that high-altitude, cold glaciers may be more responsive than traditionally assumed (Cuffey and Paterson, 2010)

## 6.7 Uncertainties

The data used in this study presented challenges and uncertainties. The largest error term in the analysis is the uncertainty

regarding radar penetration into the snowpack for the SRTM DEM. The correction we applied; a linearly scaled penetration between 0 and 5 metres above the snowline; has been widely used in the Andes. Questions remain however about the appropriateness of this approach in the semi-arid Andes where snowfall during the winter is less than in other regions. Accumulation measurements in the winters of 2010 and 2013 showed a maximum snow depth of 1.5 m and 2.0 m, respectfully (CEAZA, 2015; Kinnard et al., 2010) and some studies working in Central Chile have assumed no radar

penetration at all (Falaschi et al., 2019a; Ruiz et al., 2012). The level of radar penetration across the Andes has not been studied in detail yet and will likely result in over- or underestimations of geodetic mass balances in regions with little precipitation.

The aerial photography from 1956 and 1978 was provided without the accompanying calibration reports, while the 1956 images in particular were scanned poorly. This resulted in considerable noise in the resulting DEMs, which could be reduced

enough to quantify glacier changes, was insufficient to detect changes over the surfaces of the rock glaciers.

The LiDAR survey from 2015 was conducted just after snowfall. As such, the high rates of glacier thinning detected between 2012 and 2015 are probably under-exaggerated, especially at high elevations.

## 7. Conclusion

The La Laguna catchment contains a diverse assortment of ice-debris landforms. Tapado Glacier, which had an
approximately net-zero geodetic balance between 1956 and 1978 (-0.04 ± 0.08 m w.e. a⁻¹), has loss mass continuously between 1978 and 2020, likely explained by an increase of air temperature and a decrease of precipitation and humidity since the 1980's. The droughts that occurred within the region between 2010 and 2015 are reflected by a particularly negative geodetic balance between 2012 and 2015 (-0.54 ± 0.10 m w.e. a⁻¹), while the snow heavy winters of 2017 and 2018 resulted in a less negative mass balance between 2015 and 2020 (-0.32 ± 0.08 m w.e. a⁻¹) compared to the previous years. The ice
losses observed at Tapado Glacier conform with the regional average for the semi-arid Andes between 2000 and 2018 as well as other glaciers in the region that have been studied since the 1950s. Additionally, our study found heterogenous rock glacier behaviours and changes within the La Laguna catchment, with some rock glaciers essentially inactive whilst others are deforming at rates in excess of 4 m a⁻¹. There are also heterogenous patterns of surface elevation change both within between landforms and within individual landforms which may be indicative melting of ice lenses and the melting of internal
ice. Such analyses provide useful information on which landforms are likely of the most hydrological importance and should be studied in detail using geophysical techniques.

**Acknowledgments**

We acknowledge the Chilean Transparency Agency and the Water Directorate of Chile (DGA) for facilitating the LiDAR data from Tapado area. This work was supported by ANID + Concurso de Fortalecimiento al Desarrollo Científico de
Centros Regionales 2020-R20F0008-CEAZA, and Álvaro Ayala was supported by ANID + FONDECYT 3190732. The DGA is also thanked for the in-situ data collected with CEAZA. The authors wish to thank the European Space Agency for the free provision of the SPOT and Pléiades imagery through the Restrained Dataset project 41743, and NASA for the SRTM elevation model. We also thank many members of the CEAZA glaciology group that have collected the dGPS datasets.

**Author contributions**

BAR and SM derived the study. BAR completed the analysis with assistance from TB and SM. AA performed the climatological analysis. All authors contributed to editing the manuscript.



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
