# Peer review of "Glacier and Rock Glacier changes since the 1950s in the La Laguna catchment, Chile"

_The Cryosphere, 2021_

## Referee Comment (RC1)

Review of manuscript **'*Glacier and Rock Glacier changes since the 1950s in the La Laguna catchment, Chile*'** submitted to **The Cryosphere**

**General comments**

This manuscript presents a detailed analysis of glacier changes in Chile's La Laguna catchment based on multi-source remote sensing observations. Compared to previous studies quantifying regional glacier mass balance in the Andes of Chile, this study showed finer details of the rock glacier dynamics and temporal variations of glacier mass changes in a catchment of interest. The methods and results are clear, with quantifications compared with in-situ data and estimation from previous studies. The authors thoroughly discussed the implications of heterogenous rock glacier changes and explained the glacier response to climatological context in a long-term time frame. This study contributes to a better understanding of the underlying dynamics of rock glaciers in contrast to the intensified thinning of the typical glaciers (clean-ice/debris-covered), and an in-depth discussion of the evolution of glacial behaviors and associated landforms under the scenario of climate change. In general, the reviewer suggests modifying a few points as specified below to improve the clarity and rigor of the argument, in addition to some minor mistakes/unclarities listed in the specific comments.

(1) Uncertainty of glacier mass balance
The authors followed a method similar to Falaschi et al. (2019b?) to quantify glacier elevation change and mass balance errors. However, it is unclear how to compute the error of penetration depth (Er) in equation (4). The accounting of penetration error as an independent source may be questionable in equation (4). Given that the error of penetration depth affects the calculation of elevation changes which are then propagated to the error of mass changes, Er is not independent from $E_{\Delta v}$ in this case.

(2) Comparison with the latest regional glacier mass balance estimation
The glacier mass change estimation in this study was compared to that of Braun et al., 2019 and Dussaillant et al., 2019, which used different sources of DEMs. A new global estimation of glacier mass balance (and elevation change maps) is published in Hugonnet et al. (2021). It is necessary to update the comparisons with this dataset to see whether the disagreement persists.

(3) Comparison with in-situ glacier mass balance
In line 350, the figure (Figure 8, comparison of glacier velocities) does not match the contents about comparing with in-situ glacier mass balance. Quantitative information from the field survey is therefore missing.

(4) Discussion on the elevation changes of rock glaciers in contrast to the thinning of Tapado Glacier
Rock glaciers seem to be in an overall equilibrium (Table 5) between 2012 and 2020 in contrast to the noticeable thinning of Tapado Glacier with debris-covered and clean-ice sections (Table 4). In addition to velocities and evident elevation changes on different parts of rock glaciers, any extended comments or discussions regarding the overall state of rock glaciers? I.e., is the equilibrium state indicative of the insensitive response of glaciers to climate forcing?

**Specific comments**
Line 95: Please simply describe the annual temperature level and precipitation amount in the study region in this paragraph.

Line 160: 'Third order polynomials were fitted to elevation biases...'. According to Figure 2, six-order polynomials was used for across-track correction?

Line 195: 'We opted to follow the same methods as Falaschi et al. (2019) who utilized...' The reference is unclear, Falaschi et al. (2019a) or Falaschi et al. (2019b)?

Line 251: When describing glacier area changes, keep the number (positive/negative) consistent to avoid confusion. The sentence can be revised to '...the glacier area decreased at a rate of 5910 ±1060 $m^2$ $a^{-1}$ (0.35 ± 0.30 % $a^{-1}$), which increased to 6818 ± 24202 $m^2$ $a^{-1}$ (0.60 ± 2.28 % $a^{-1}$)...'

Line 251 Page 12: '-5910 ±1060 $m^2$ $a^{-1}$ (-0.35 ± 0.30 %$a^{-1}$)', missing space between units (% $a^{-1}$). This kind of error is widely found throughout the manuscript (i.e., line 256, 315, 316). Do proofreading and correct the missing or surplus spaces.

Line 265: '()' missing references?

Line 278: '...between 2012 to 2015' revise to 'from 2012 to 2015'

Tables: The format of tables (number format, border lines, etc.) needs to be revised to improve the reading and be in line with the journal's requirements.

Table 5: The table is long, moving to the appendix or supplementary?

Figures:

Figure 1: It is not clear about the extent of debris-covered sections in (b). This information is necessary for a better interpretation of Figure 5. Try set the shade of rock glacier extent more contrasted in (a). The location of the (c) is not described in the figure caption. For a concise presentation, (b) and (c) can be aligned horizontally rather than vertically with (a). This organization also applies to Figure 3.

Figure 4: The legend covers up (blurs) part of the line drawings.

Figure 8: The figure does not match the contents discussed. It is about the validation of glacier velocities rather than glacier surface elevation changes from the field survey.

Figure 11: To improve visual geolocation, set the scale of the same place consistent across different panels (a, b, c).

References:

Hugonnet, R., McNabb, R., Berthier, E., Menounos, B., Nuth, C., Girod, L., Farinotti, D., Huss, M., Dussaillant, I., Brun, F., and Kääb, A.: Accelerated global glacier mass loss in the early twenty-first century, Nature, 592, 726–731, https://doi.org/10.1038/s41586-021-03436-z, 2021.

---

## Referee Comment (RC2)

Review of Glacier and Rock Glacier changes since the 1950s in the La Laguna catchment, Chile
by Benjamin Aubrey Robson et al.

I found that the methods are appropriate and the discussion and conclusions that are drawn are appropriate and relevant for the readership of The Cryosphere. There are some minor corrections to the text that should be made before publication, but overall I recommend that this work be published.

Detailed comments:

line 14: consider 'rapidly' instead of 'strongly'

line 15: 'less investigated' is comparative please add to this sentence to complete it.

Line 19: it is unclear which glacier you are referring to at this point.

Line 21: suggest that 'strong' not be used as a modifier as it is a bit loose.

Line 23: remove 'in the region' as it is repetitive from the last sentence.

Line 35: Better if this paragraph was a bit longer.

Line 40-42: It is not clear why this needs to be an either or situation.

Line 45: would add an e.g, here as the citation list is not exhaustive

Line 55: The citations are not in the correct order starting with oldest and progressing to youngest. Correction should be applied throughout.

Line 82. maybe use 'agricultural production' 'worth' instead of 'of'. Also correct typo associated with 'industry'

Figure 1. The blue stars are hard to read in this figure and the red stars overlap. Consider a white outline of the markers? Caption: Correct typos 'Location of the La Laguna catchment and of landforms studies in this paper. Three landforms that will be discussed'
in the text are highlighted in (A).

Table 1: 'For the imagery that were used to produce DEMs, the RMSE values for the Ground Control Points and Tie Points are shown.' You can remove 'that were'

Line 109: 'The data *were*"

Line 127: 'over' to 'on'

Line 132: can cut 'for this area'

Table 2. Would be helpful to signify in the figure by the bottom row is in italics.

Figure 3. It is difficult to read these lines.

Figure 4. y-axis needs a label.
Line 351. Remove 'as'

Figure 8.  A-A' (etc.) should be labeled in Figure 1.

Line 398. need a '.' here.

Line 448. remove 'as'

Line 467. Additionally Anderson et al., 2018 point simply to climate warming as a cause of the transition between debris-covered and rock glaciers.

Line 496. would be helpful to cite who has used this approach in the past.

References

Anderson, R. S., Anderson, L. S., Armstrong, W. H., Rossi, M. W., and Crump, S. E.: Glaciation of alpine valleys: The glacier – debris-covered glacier – rock glacier continuum, 311, 127–142, https://doi.org/10.1016/j.geomorph.2018.03.015, 2018.

---

## Author Response (AR1)

**Response to Reviewers**

Dear reviewers,

We would like to thank you for taking the time to review our manuscript and providing constructing and helpful feedback. Please find our responses in the tables below.

Reviewer 1

| Reviewer comment | Our response |
|---|---|
| The authors followed a method similar to Falaschi et al. (2019b?) to quantify glacier elevation change and mass balance errors. However, it is unclear how to compute the error of penetration depth (Er) in equation (4). The accounting of penetration error as an independent source may be questionable in equation (4). Given that the error of penetration depth affects the calculation of elevation changes which are then propagated to the error of mass changes, Er is not independent from $E\Delta v$ in this case. | We have now clarified in L 188 – 191 that a linear correction was applied, where 0 m correction was applied at the firn line, increasing to 5 m correction at the top of the glacier. The reason we have Er as a separate error term is that it is only included in the DEM pairs that involve the SRTM DEM. We recognise that there are various different corrections possible for radar penetration, and we have tried to list them in the text. The reason we chose this method is that it had been applied in other South American studies. We hope that by incorporating the radar correction into $E\Delta v$ we can present our results and demonstrate their significance. We have also clarified now that we follow the approach of Falaschi et al 2019b. |
| The glacier mass change estimation in this study was compared to that of Braun et al., 2019 and Dussaillant et al., 2019, which used different sources of DEMs. A new global estimation of glacier mass balance (and elevation change maps) is published in Hugonnet et al. (2021). It is necessary to update the comparisons with this dataset to see whether the disagreement persists. | Thank you for the update, we have now compared our study to that of Hugonet et al, 2021 both in the text and the figure. |
| In line 350, the figure (Figure 8, comparison of glacier velocities) does not match the contents about comparing with in-situ glacier mass balance. Quantitative information from the field survey is therefore missing. | We have updated the text to refer to figure 7. We also calculated the median deviation, which is substantial, and as such validates our interpretation that the relation is weak. |
| Rock glaciers seem to be in an overall equilibrium (Table 5) between 2012 and 2020 in contrast to the noticeable thinning of Tapado Glacier with debris-covered and clean-ice sections (Table 4). In addition to velocities and evident elevation changes on different parts of rock glaciers, any extended comments or discussions regarding the overall state of rock glaciers? I.e., is the equilibrium state indicative of the insensitive response of glaciers to climate forcing? | This is a good point. We have expanded on this point in the discussion. It is hard to compare a glacier surface that is stable with a stable rock glacier surface. The former indicates a mass balance close to zero, but the latter can indicate either that the rock glacier is in equilibrium or conversely that there is little permafrost to thaw in the rock glacier. We have emphasis this point more in the manuscript and we now suggest that surface elevation changes combined with ice rock glacier deformation rates is the best |

| | way of assessing if rock glaciers have lost ice. |
|---|---|
| Line 95: Please simply describe the annual temperature level and precipitation amount in the study region in this paragraph. | We have now added in this information |
| Line 160: 'Third order polynomials were fitted to elevation biases...'. According to Figure 2, six-order polynomials was used for across-track correction? | Thanks for spotting this. You are right, sixth-order polynomials were useds for along-track, and third order for across track and elevation dependent. This has now been fixed. |
| Line 195: 'We opted to follow the same methods as Falaschi et al. (2019) who utilized…' The reference is unclear, Falaschi et al. (2019a) or Falaschi et al. (2019b)? | This has now been fixed. |
| Line 251: When describing glacier area changes, keep the number (positive/negative) consistent to avoid confusion. The sentence can be revised to '…the glacier area decreased at a rate of 5910 ±1060 m2 a -1 (0.35 ± 0.30 % a-1 ), which increased to 6818 ± 24202 m2 a -1 (0.60 ± 2.28 % a-1 )…' | This has been revised |
| Line 251 Page 12: '-5910 ±1060 m2 a -1 (-0.35 ± 0.30 %a-1 )', missing space between units (% a-1 ). This kind of error is widely found throughout the manuscript (i.e., line 256, 315, 316). Do proofreading and correct the missing or surplus spaces. | Thanks, we have now fixed this |
| Line 265: '()' missing references? | Thanks, we have now fixed this |
| Line 278: '…between 2012 to 2015' revise to 'from 2012 to 2015' | This has now been fixed |
| Tables: The format of tables (number format, border lines, etc.) needs to be revised to improve the reading and be in line with the journal's requirements | This has now been fixed |
| Table 5: The table is long, moving to the appendix or supplementary? | This has now been fixed |
| Figure 1: It is not clear about the extent of debris-covered sections in (b). This information is necessary for a better interpretation of Figure 5. Try set the shade of rock glacier extent more contrasted in (a). The location of the (c) is not described in the figure caption. For a concise presentation, (b) and (c) can be aligned horizontally rather than vertically with (a). This organization also applies to Figure 3. | Thank you for the feedback. We have now altered the symbology for the rock glaciers, debris-covered section, and clean ice. We have changed the orientation of Figure 1 and Figure 3. |
| Figure 4: The legend covers up (blurs) part of the line drawings. | This has now been fixed |
| Figure 8: The figure does not match the contents discussed. It is about the validation of glacier velocities rather than glacier | We will move the figure up so that it fits better in the text |

| surface elevation changes from the field survey | |
|---|---|
| Figure 11: To improve visual geolocation, set the scale of the same place consistent across different panels (a, b, c) | We have now made the scale consistent |

Reviewer 2

| Reviewer comment | Our response |
|---|---|
| line 14: consider 'rapidly' instead of 'strongly' | This has now been changed |
| line 15: 'less investigated' is comparative please add to this sentence to complete it. | This has now been changed |
| Line 19: it is unclear which glacier you are referring to at this point. | We now make it clear that we are referring to Tapado Glacier |
| Line 21: suggest that 'strong' not be used as a modifier as it is a bit loose. | We have changed 'strong' to 'increased' |
| Line 23: remove 'in the region' as it is repetitive from the last sentence. | This has now been changed |
| Line 35: Better if this paragraph was a bit longer. | Thanks for the suggestion, we have added a sentence |
| Line 40-42: It is not clear why this needs to be an either or situation. | We agree that a combination of both theories for rock glacier formation is most likely, but here we just present both schools of thought from the literature. |
| Line 45: would add an e.g, here as the citation list is not exhaustive | This has now been changed |
| Line 55: The citations are not in the correct order starting with oldest and progressing to youngest. Correction should be applied throughout. | Thanks for pointing this out. We have corrected it |
| Line 82. maybe use 'agricultural production' 'worth' instead of 'of'. Also correct typo associated with | This has now been changed |
| Figure 1. The blue stars are hard to read in this figure and the red stars overlap. Consider a white outline of the markers? Caption: Correct typos 'Location of the La Laguna catchment and of landforms studies in this paper. Three landforms that will be discussed' in the text are highlighted in (A). | We have substantially edited the symbology of this figure so hope it is more readable now. Unfortunately at this scale the red markers do overlap, this is because the location of the mass balance data is approximately the same from year to year, but has a slight shift. |
| Table 1: 'For the imagery that were used to produce DEMs, the RMSE values for the Ground Control Points and Tie Points are shown.' You can remove 'that were' | This has now been changed |
| Line 109: 'The data were" | This has now been changed |
| Line 127: 'over' to 'on' | This has now been changed |
| Line 132: can cut 'for this area' | This has now been changed |
| Table 2. Would be helpful to signify in the figure by the bottom row is in italics. | We have removed the italics |
| Figure 3. It is difficult to read these lines. | This has now been changed |
| Figure 4. y-axis needs a label. | Thanks for spotting this, this has now been fixed |
| Line 351. Remove 'as' | This has now been changed |
| Figure 8. A-A' (etc.) should be labeled in Figure 1. | We have added these laels |
| Line 398. need a '.' here. | This has now been changed |
| Line 448. remove 'as' | This has now been changed |
| Line 467. Additionally Anderson et al., 2018 point simply to climate warming as a cause | We have adapted this sentence and added this reference |

| | |
|---|---|
| of the transition between debris-covered and rock glaciers | |
| Line 496. would be helpful to cite who has used this approach in the past. | We have added in some references here |